# Plantar Fasciitis in Soccer Players—A Systemic Review

**DOI:** 10.3390/ijerph192114426

**Published:** 2022-11-03

**Authors:** David C. Noriega, Ángel Cristo, Alejandro León, Belén García-Medrano, Alberto Caballero-García, Alfredo Córdova-Martinez

**Affiliations:** 1Department of Orthopedic Surgery, Clinic University Hospital of Valladolid, 47005 Valladolid, Spain; 2Department of Surgery, Ophthalmology, Otorhinolaryngology and Physiotherapy, Faculty of Medicine, University of Valladolid, 47005 Valladolid, Spain; 3Department of Anatomy and Radiology, Faculty of Health Sciences, GIR Physical Exercise and Aging University of Valladolid, University Campus Los Pajaritos, 42004 Soria, Spain; 4Biochemistry, Molecular Biology and Physiology, Faculty of Health Sciences, GIR Physical Exercise and Aging, Campus Duques de Soria, University of Valladolid, 42004 Soria, Spain

**Keywords:** sport, soccer, foot, biomechanics, plantar fasciitis, systematic review

## Abstract

Soccer is one of the most popular sports in the world. Players often suffer a variety of injuries, the most common being injuries to muscles and tendons. It is striking that with soccer, being the most practiced sport, and considering that most injuries occur in the lower extremities, plantar fasciitis (PF) is not one of the most frequent injuries (at least in terms of clinical data collected). The purpose of this review was to provide a comprehensive update of the topic “plantar fasciitis” focusing on soccer players. The review was conducted in accordance with the PRISMA (Preferred Reportiog ltems for Systmiatic reviews and Meta-Analyses) statement. PubMed, Cochrane Library and Scopus were researched. PICO (Patient, Population or Problem; Intervention; Comparison; and Outcome) components were identified. The keywords used were “plantar fasciitis”, “plantar fasciitis and sport”, “plantar fasciitis risk factors”, “plantar fasciitis soccer” and “plantar fasciitis football players”. With respect to the objective proposed for the research, we found eight specific articles focused on soccer. Of these, five were general reviews discussing the different methods of treatment of this pathology, and we have only found three studies that focused on PF in soccer, with two of them referring to a clinical case whereby the report and discussion only dealt with the specific treatment followed by the soccer player. After reviewing the manuscripts included in this work, we were surprised that there is no data in which the Silfverskiöld test was performed, as this test explores the passive mobility of the ankle and the degree of dorsiflexion in the supine position. We concluded that soccer players suffer pain in the sole of the foot compatible with plantar fasciitis; however, as indicated by Suzue et al., it is often not diagnosed because the athlete does not consider performing the clinical examinations necessary for its diagnosis. The shortage of reported publications in soccer may mask other PF-associated injuries.

## 1. Introduction

Players often suffer a variety of injuries, the most common being those affecting muscles and tendons. The anatomical regions where most injuries occur are the knee joint, ankle, thigh, groin, hip and foot. The plantar fascia is a structure made up of fibrous and dense connective tissue, which is located in the medial tuberosity of the calcaneus and the metatarsophalangeal joints of each toe. It can be divided in three portions, called the medial, lateral and central band, of which the latter is usually the largest, and fulfills important functions within the biomechanics of the foot as it is the support for the longitudinal plantar arch of the foot [1,2] (Figure 1).

When walking, 100% of the body weight rests on the heel during the first rocker of the gait, causing tension in the plantar fascia. This simple fact, together with other influences, is connected to the appearance of plantar fasciitis (PF). Many authors have studied the predisposing risk circumstances for plantar fasciitis [1,2,3,4,5,6,7].

PF is associated with a variety of sports, but it is most commonly reported in recreational and elite runners (incidence 5% to 10%) [1]. Among the risk factors associated with this musculoskeletal pathology are biomechanical abnormalities of the foot, sedentary lifestyle, obesity, prolonged standing, improper exercise, overtraining, limited ankle dorsiflexion motion, etc. Petraglia et al. [1] divide the causes into intrinsic (anatomical, functional and degenerative) and extrinsic (due to overuse, incorrect training and inadequate footwear). In each of these sections they include the different risk factors. Other authors have also expressed the same semiology [6,8,9].

In cadaver studies, it has been seen that there is a continuity of the plantar fascia with the paratendon of the Achilles tendon, providing a positive correlation between the load of the Achilles tendon and the tension of the plantar fascia [10].

### 1.1. Plantar Fascia and the Windlass Mechanism

From a biomechanical point of view, the plantar fascia represents one of the most important structures for preserving the integrity of the internal longitudinal arch (ILA) of the foot, and thus maintaining the plantar vault, together with the plantar long ligament and Spring’s ligament (calcaneal-astragalus-scaphoid ligament), assisting the intrinsic musculature. The fascia supports and keeps the plantar arch [11]. 

Ker et al. [12] have indicated that the fascia acts as an energy storage device in the foot, acting as a cushion against the forces that appear in the toe-off phase of gait, creating a framework under the metatarsal heads thanks to the tension of the soft tissues.

During forward gait phases or rockers, the heel contacts the ground and performs a small flexion of the ankle and the metatarsophalangeal joint of the big toe thanks to the action of the extensor hallucis longus of the first toe. With the forefoot or third rocker, this extension tightens the plantar fascia bringing the two pillars of arch support closer together. It prompts the metatarsophalangeal joint to verticalize, elevating the plantar arch and supinating the rearfoot at the same time [11,12,13].

When the foot contacts the ground, in the phase of maximum contact or monopodal stance, the weight of the body is transmitted to the plantar vault which flattens, and then the plantar arch descends and the calcaneus becomes horizontal (dorsiflexed rearfoot). At the same time, a plantar flexion of the metatarsophalangeal joint occurs, tensing the plantar fascia [11,13]. When the motor impulse is then produced, the heel must ascend by the action of the ankle extensors, and the plantar fascia tightens and raises again with the extension of the metatarsophalangeal. In this phase, the first metatarsal is verticalized, and the fascia increases in tension since its origins move away, causing the arch to stabilize [11,13].

Thus, any abnormality of the Windlass mechanism will affect any related area, mainly the plantar fascia (also the Achilles Tendon). The most important causes of Windlass mechanism dysfunction are due to improper function of the first toe, referred to as functional hallux limitus (FHL). This is defined as a limitation of dorsal flexion of the first metatarsophalangeal joint during the propulsive phase of gait, with no limitation in unloaded biomechanical conditions [14]. 

During the toe-off of the first toe from the ground in gait, it must bear a load equivalent to two or three times the weight of the person, so there is an enormous pressure that it must withstand at this instant [15]. On the other hand, the decrease in ankle dorsiflexion is a risk factor for developing fasciitis. Decreased ankle mobility may be due to restriction of joint tissues involved in the movement [16].

To carry out the assessment of ankle influence, the Silfverskiöld test is performed. This test that explores the degree of ankle dorsiflexion in the supine position in two different positions: with the knee extended and flexed [17]. The test is used to evaluate the contracture of the ankle joint whether it is caused by gastrocnemius muscle contracture or by the Achilles tendon contracture.

Therefore, during walking, jumping and running, there are forces that stress the foot and alter the internal longitudinal arch (ILA). A good orientation of the plantar fascia is important to allow the foot to settle well to the ground, to help control pronation and supination of the foot and to stabilize the arch. If the function of the ILA is not correct, the stress on the fascial tissue increases. In addition, it is able to distribute the weight exerted on the foot between all the metatarsal heads. The plantar fascia also provides greater efficiency to propulsive forces, twice as much when running, being a cushioning mechanism for the soft tissues underneath the metatarsal heads in the late plantar stance phase [18,19,20].

### 1.2. Etiopathogenesis of Plantar Fasciitis

PF generates heel pain. The pain is usually caused by the collagen degeneration process (sometimes mistakenly referred to as “chronic inflammation”) at its origin in the medial tubercle of the calcaneus [21]. 

This degeneration is similar to chronic necrosis of tendinosis, which is characterized by loss of collagen continuity, increased ground substance (connective tissue matrix) and vascularity and the presence of fibroblasts rather than the inflammatory cells usually seen in acute tendinitis [22]. The cause of degeneration is repetitive microtears of the plantar fascia that exceed the body’s ability to repair them by itself [4].

During this progressive event, an increase in the number of fibroblasts is developed and added to the fragmentation of the ground substance, a myxoid degeneration (accumulation of acid mucopolysaccharides in the connective tissue with alteration of the fibrillar elements) and neovascularization. The entire procedure is attributed to the normal repair capacity of the tissues being exceeded. Tissue fatigue can also be observed due to excess traction, degeneration and micro tears in the collagen tissue [23,24].

Secondary inflammation is caused by repeated microtrauma to the medial calcaneal tuberosity, which can lead to bony degenerative changes in and generate periostitis of the medial calcaneal tubercle. This can lead to the finding of calcification and subsequent development of a calcaneal spur [25,26]. 

### 1.3. Plantar Fasciitis in Sport

PF occurs in both sedentary individuals and athletes of all ages. However, plantar fasciopathy occurs most frequently in women aged 40–60 years [27,28,29,30]. Furthermore, PF occurs in both recreational and elite athletes and has been described in different sports [31]. In athletes, it is particularly frequent in those who perform running and dancing activities that require maximum plantar flexion of the ankle and dorsiflexion of the metatarsophalangeal joint [25,26,27,28,29,30]. In many cases, due to its painful component, it has a significant impact on the sporting and occupational environment. 

In soccer players and runners, Achilles tendinopathies are the most frequent injury. In addition, in basketball players and runners, the prevalence of a stress fracture or PF is higher [32]. These data coincide with those detected in elite athletes who competed in the London 2012 Olympic Games [33]. 

## 2. Objectives

It is striking that soccer, being one of the most practiced sports (according to federative licenses) and considering that the lower extremities are the most injured regions (knee, feet and ankle), PF is not one of the most frequent injuries in that sport. This fact, moreover, coincides with the scarce number of scientific publications on PF in soccer players. For this reason, in this systematic review, we have carried out a study on PF in soccer, contemplating the most relevant aspects of this pathology.

## 3. Methodology

The review was conducted in accordance with the PRISMA statement. PubMed, Cochrane Library and Scopus were researched. PICO (Patient, Population or Problem; Intervention; Comparison; and Outcome) components were identified. The keywords used were “plantar fasciitis”, “plantar fasciitis and sport”, “plantar fasciitis risk factors”, “plantar fasciitis soccer” and “plantar fasciitis football players”. Each search term was mapped to specific MeSH subject headings within each database (Figure 2). The reference lists of plantar fasciitis systematic reviews and other studies were manually checked. After examining the titles and abstracts of the identified articles, they were compared to the lists of included and excluded papers.

### 3.1. Inclusion Criteria

All systematic reviews or meta-analyses linked to the topic of plantar fasciitis (e.g., risk factors, diagnosis, or treatments) were eligible for inclusion. Reviews on plantar fasciopathy or fasciosis were also included because these terms used to be interchangeable with plantar fasciitis. Something similar happens with plantar heel pain, therefore some studies were included whereby authors did not create a strict distinction with plantar fasciitis.

### 3.2. Exclusion Criteria

Articles concerning military personnel and subjects with systemic diseases in addition to PF were excluded. Although conference abstracts and conference papers were reviewed during the research, they were all considered inadequate due to the paucity of data related to the study design and intervention program. The articles not published in English and abstracts were also excluded because it was difficult to assess the methodological quality of those systematic reviews.

## 4. Results

With respect to the objective proposed for the research, we found eight specific articles focused on soccer. Of these, five were general reviews informing about the different methods of treatment of this pathology (not centered on soccer), and we only found three studies focused on PF in soccer, with two of them referring to a clinical case whereby the report and discussion only dealt with the specific treatment followed by the player. We found only one study in which the authors [27] investigated the prevalence and characteristics of lower heel pain in 1473 professional, semi-professional and amateur soccer players. The authors [34] concluded that lower heel pain among soccer players was primarily caused by local biomechanical stress, although they also found other causes such as sural nerve compression [34].

On the other hand, Suzue et al. [35] have expounded on plantar fascia rupture. They indicated that in the case of plantar fasciitis, it is a very exceptional situation and its approach should be different from the point of view of symptomatology, diagnosis and treatment.

Costa and Dyson [36] presented a case of a 15-year-old female, who presented with symptomatic chronic plantar fasciitis. The study focuses on the integrative use of acetic acid iontophoresis in combination with active rehabilitation for her treatment

Therefore, although soccer is one of the most popular team sports in the world, PF is not a common pathology. When it occurs, it is usually caused by overload, which is secondary to any of the causes that we have already indicated as risk factors for this pathology.

## 5. Discussion

The purpose of this review was to provide a comprehensive update of the topic “plantar fasciitis” focusing on soccer players. A total of 1808 articles were initially identified and 96 were revised according to the inclusion criteria. Surprisingly, concerning soccer, there was an almost nonexistent number of articles. Although we suspect that many soccer players suffer from this pathology, we believe they are not specifically treated and therefore no conclusive studies have been carried out in this area.

As numerous authors published before, PF is fundamentally due to an overload that is accompanied by an inflammatory process. Essentially, we can have concluded that PF is a pathology characterized by inferomedial heel pain, accompanied by inflammation, although plantar fasciitis is not a primary acute phase process [2,23,37]. As mentioned above, it occurs when the plantar fascia is subjected to overly tensional and mechanical loads, which can generate microtears that trigger an inflammatory repair process [2,25,36].

The foot biomechanical structure is directly related to the pathology of plantar fasciitis, given its function of absorbing impact, adapting to the ground, and supporting the body where the plantar fascia, with its windlass mechanism [13], helps to control arch stability, in addition to control the degree of pronation or supination of the foot during the rockers of gait. Any condition that increases the tension of the fascia will trigger this pathology [13].

As reported by Suzue et al. [35], a large number of soccer players suffered from heel pain, but only 26.9% of them decided to undergo a scan. The cause of injuries is multifactorial and is related to repetitive microtrauma caused by impact forces in technical movements, but also by the playing field, the mode of exercise, recovery time, climatic conditions and the footwear used [38].

However, as indicated by Bolgla and Malone [13], dorsiflexion limitation is an ethological factor of plantar fasciitis as this ankle movement is necessary during the gait cycle, allowing the foot to advance and overtake the other foot during a step. People who cannot perform adequate dorsiflexion compensate by unblocking the midtarsal joints; this produces excessive pronation of the foot and overloads the plantar fascia [13].

In our previous studies [5], we reported that a sport such as skiing could alleviate and be helpful in the treatment of plantar fasciitis. In skiing, the ability to control body position is limited by boots that restrict the mobility of the ankle joint and force a change of body inclination from neutral to lean [39]. In control over the sagittal plane, the boots have about 17° of anterior tilt over the vertical; that is, the ankle joint remains at 17°. There is an elongation of the muscle fibers of the triceps suralis and, consequently, an upward tension on the calcaneus. That situation creates an elongation tension on the plantar fascia, facilitating the stretching of the plantar fascia, and therefore its recovery.

It is striking that none of the articles reviewed proceeded to perform an assessment of ankle biomechanics. This joint is key in striking the ball when heading it, and in spite of this, there is no data in which the Silfverskiöld test is performed [17]. However, this test is not only used for soccer players, but also for other sports. This test explores the passive mobility of the ankle and the degree of dorsiflexion in the supine position. This maneuver looks for possible abnormalities in the retraction of the gastrocnemius muscle or the entire triceps suralis. In the restriction of the gastro-soleus complex or in a joint injury, it is considered positive if there is limitation in dorsiflexion with the knee flexed and extended [17]. Decreased ankle mobility may be due to biarticular (e.g., calf) or monoarticular (e.g., soleus) muscle tissue restriction or joint problems [16,17]. García-Vidal et al. [17] stated that subjects with ankle dorsiflexion limitation associated with calf restriction are more likely to suffer from PF. According to Kibler et al. [25], flexor muscle flexibility deficits may contribute to increased fascial stretch. Cheung et al. indicated that intense muscle contractions of the plantar flexor muscles cause indirect stretching of the fascia, increasing the risk of developing PF [40].

Based on our clinical experience, both in the normal population and in other sports (mainly runners on foot), the plantar fascia tends to lose elasticity. To this we could add that we have also observed that it is more frequent in people who do less exercise, and more in those who do not stretch the fascia, a fact that we have already published [5] and which shows that this situation is the opposite of the preventive mechanism. 

In soccer players, the most common technical movements are tackling, being tackled, sprinting, running, turning, jumping, landing and kicking the ball. These are all complex movement patterns that require a high degree of coordination and body control. The biomechanics and explosiveness of the stride, as well as the hyperextension of the ankle in the kicking mechanics generate a risk situation for the plantar fascia that could lead to its injury. In this regard, Laurie et al. showed that the comparison of different studies is difficult due to the different classification systems used to indicate the severity of the injury; some studies are classified according to the medical treatment used, whereas others are classified by the days of absence from competition [41].

In soccer, there are also other risk factors for PF-related injuries, such as age, gender, skill level, environment and surface type. With increasing age, the fat pad atrophies, which impedes the proper shock absorption function of the forces that impact the heel [42].

## 6. Conclusions

In conclusion, soccer players, similar to other athletes, suffer pain compatible with that of plantar fasciitis. As indicated by Suzue et al. [35], it is often infra-diagnosed because the athlete does not consider performing the clinical examinations necessary. On the other hand, and from the technical point of view, currently, several authors have spoken about “plantar fasciosis”, which means “degeneration” of the plantar fascia, because, although it is accompanied by inflammation, an acute process is not the origin of this pathology. The shortage of reported publications in soccer may mask other PF-associated injuries.

## Figures and Tables

**Figure 1 ijerph-19-14426-f001:**
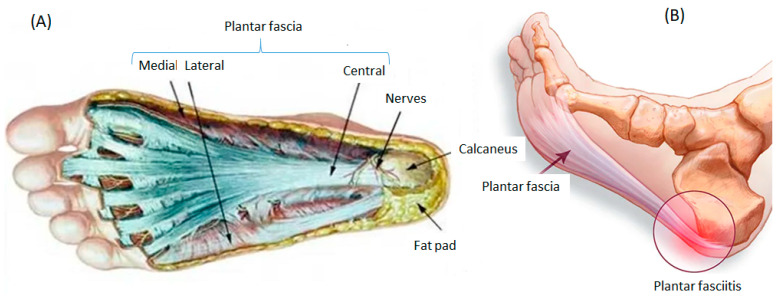
Plantar fascia and plantar fasciitis (PF): (**A**) Frontal plane of the plantar fascia with its different types insertions; (**B**) Posterior insertion of plantar fascia in calcaneus.

**Figure 2 ijerph-19-14426-f002:**
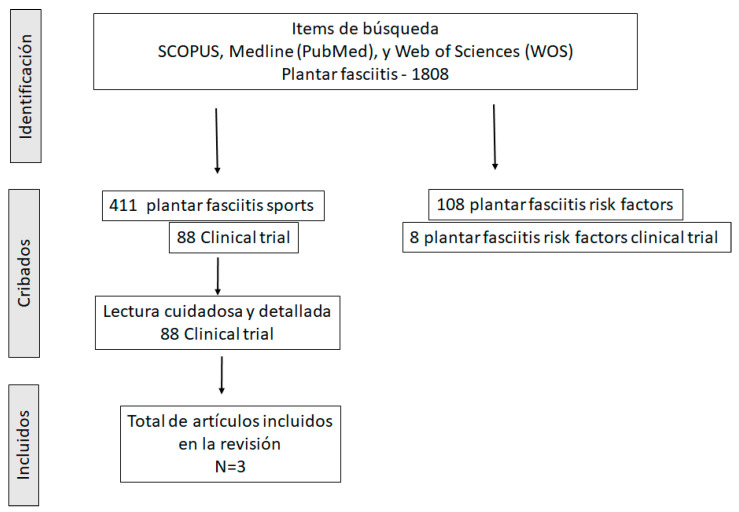
Studies selected for developing the systematic review of plantar fasciitis in soccer players.

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
