# Peer review of "Plantar Fasciitis in Soccer Players—A Systemic Review"

_ijerph, 2022, doi:10.3390/ijerph192114426_

Round 1
Reviewer 1 Report
General comments:
Plantar Fasciitis (PF) is one of the causes of pain in the heels of athletes. The authors have presented a review of PF and extensively searched the literature using certain keywords from public databases for finding studies related to Soccer players. The authors have done a great job in identifying the relevant articles from a list of 1,808. It was found that there is a limited number of studies on soccer players and their injuries relayed to PF.
Major comments:
One of the major concerns is the novelty of this study and its conclusion. It was found that a similar study has been done previously (PMID: 28717618) and this article looks similar to that. The figure that gives a flowchart of article selection is almost similar to the one in PMID 28717618. Also, the findings of this study i.e. description of 3 papers in the result section could have been expanded to show the correlation between PF and soccer injuries. Overall, the organization of the paper could have been better. There are several grammatical errors, strikethrough letters, clipped figures, figures without figure numbers, legend, etc. The overall conclusion of this review paper is that there are not enough studies on PF and soccer injuries. It was not clear whether it is really a gap in these areas of study. If yes, the authors could have explored a little bit more about why there is a lack of publishing such studies. It is understandable that the article was written by non-native English speakers, so it would be really nice if the authors can take some help in this regard.
Minor comments:
There are several mistakes that caught my attention and here are some of them.
-
Some words like “fasciitis” are written as “fascitis”. It is understandable that the meaning may be the same in two different languages, but it should be good to be consistent.
-
The manuscript contains several grammatical errors. Some examples are
-
The sentence that starts at Line 94: “This test that explores …”
-
Line 96: Rewrite the sentence; what is ALI?
-
Strikethrough texts found in the manuscript
-
PRISMA abbreviation should be expanded before first use
-
Missing text, Figure legend, number (2nd Figure, flow chart)
Author Response
Dear Reviewer,
Thank you for taking the time to read the manuscript and give us valuable comments. We tried to address all of them. We hope, that the corrected version meets your requirements and will be accepted for publication
General comments:
Plantar Fasciitis (PF) is one of the causes of pain in the heels of athletes. The authors have presented a review of PF and extensively searched the literature using certain keywords from public databases for finding studies related to Soccer players. The authors have done a great job in identifying the relevant articles from a list of 1,808. It was found that there is a limited number of studies on soccer players and their injuries relayed to PF.
Major comments:
1.- One of the major concerns is the novelty of this study and its conclusion. It was found that a similar study has been done previously (PMID: 28717618) and this article looks similar to that. The figure that gives a flowchart of article selection is almost similar to the one in PMID 28717618.
Thank you for this remark.
Related to this specific topic, I would like to remark to your comments, we like remark that that the authors have focused the study on athletes in general, and focused on diagnosis and treatment. The authors conclude that "we could not find any specific diagnostic algorithm for FP in athletes, because no different diagnostic strategies were used for athletes and non-athletes". Our study is specifically focusing soccer players.
Petraglia et al (reference 1 in our paper) throughout their presentation emphasise risk factors (table 1), diagnosis (table 2) and treatment (table 3).
We have studied and analysed in depth this work by Petraglia et al, and sincerely, we believe that it is a very different approach.
Regarding the figure that provides a flow chart of the selection of articles, you indicate that it is almost similar. This is not strange because the primary objective is the study of plantar fasciitis. According the Petraglia et al objectives, they have included 17 articles, however, according to our objectives, and specifically focused on soccer, there were only 3 manuscripts.
2.- Also, the findings of this study i.e. description of 3 papers in the result section could have been expanded to show the correlation between PF and soccer injuries.
Thank you for your comment. In this respect, you are right, but if we want to focus only in soccer, it does not allow us to draw other conclusions, as we have mentioned, soccer players are reluctant to communicate the pathology and therefore it would not allow us to clarify further aspects concerning its relationship with other injuries.
3.- Overall, the organization of the paper could have been better. There are several grammatical errors, strikethrough letters, clipped figures, figures without figure numbers, legend, etc.
Many thanks. Indeed, the figure was cropped and sincerely, we don't know why. Now we have reloaded the complete figure, and we have added the foot of the figure.
4.- The overall conclusion of this review paper is that there are not enough studies on PF and soccer injuries. It was not clear whether it is really a gap in these areas of study. If yes, the authors could have explored a little bit more about why there is a lack of publishing such studies.
Thank you for your comment. It is true what you say, but with such a small number of publications, it is difficult for us to extend the conclusion without speculation.
In fact, we have already said that "although we suspect that many footballers suffer from this pathology, we believe that there is no specific treatment for it and therefore no conclusive studies have been carried out". To complete more the conclusions, we have included this sentence: “The shortage of reported publications in soccer may mask other PF-associated injuries.”
5.- It is understandable that the article was written by non-native English speakers, so it would be really nice if the authors can take some help in this regard.
Thank you for this remark. We have had corrected the English by a professor from our faculty. However, if necessary, we are willing to pay for the proofreading if it is done by the publisher itself.
Minor comments:
There are several mistakes that caught my attention and here are some of them.
Some words like “fasciitis” are written as “fascitis”. It is understandable that the meaning may be the same in two different languages, but it should be good to be consistent.
We have revised the possible mistakes.
The manuscript contains several grammatical errors. Some examples are
6.- The sentence that starts at Line 94: “This test that explores …”
We have added: “The test is used to evaluate the contracture of the ankle joint whether it’s caused by gastrocnemius muscle contracture or by the Achilles tendon contracture”
7.- Line 96: Rewrite the sentence; what is ALI?
Is the “internal longitudinal arch (ILA)”. We have corrected it in the text
8.- Strikethrough texts found in the manuscript
Thanks, we have deleted
9.- PRISMA abbreviation should be expanded before first use
Preferred Reportiog ltems for Systmiatic reviews and Meta-Analyses.
Thanks, we have introduced in the text
10.- Missing text, Figure legend, number (2nd Figure, flow chart)
The text of figure has been introduced
Reviewer 2 Report
Title: PLANTAR FASCIITIS IN SOCCER PLAYERS. SYSTEMIC REVIEW - please add "A" before systematic review to make it "A systematic review"
Abstract: PF just appeared as an abbreviation without first being written in full, and abbreviation follows in a bracket. I see track changes on the abstract? The abstract of a systematic review is a summary of studies used in a systematic review.
Keywords: Please add "systematic review"
Introduction: well reviewed. I still see the some track changes.
Methods: Well described. The review has adhered the guidelines for conducting a systematic review, such as PRISMA. However, since relevant papers were nearly non-existent, I understand why the manuscript has no table that shows the primary studies with their study design, country, sample, study population, analysis and findings, etc., which is the norm in a systematic review method.
Results and discussion: fairly written considering the limited relevant studies. The authors have reported that "there were almost nonexistent number of articles. Although we suspect that many soccer players suffer from this pathology, we believe they are not specifically treated and therefore no conclusive studies have been carried out in this area". But what are the implications of non-diagnoses? That is not clear in the conclusion.
Author Response
Dear Reviewer,
Thank you for taking the time to read the manuscript and give us valuable comments. We tried to address all of them. We hope, that the corrected version meets your requirements and will be accepted for publication
Title: PLANTAR FASCIITIS IN SOCCER PLAYERS. SYSTEMIC REVIEW - please add "A" before systematic review to make it "A systematic review"
Thanks, we add the preposition
1.- Abstract: PF just appeared as an abbreviation without first being written in full, and abbreviation follows in a bracket. I see track changes on the abstract? The abstract of a systematic review is a summary of studies used in a systematic review.
Thank you very much for your appreciation. We have been reviewing the summary and we have introduced other sentences to complete it.
Sentences introduced:
"The purpose of this review was to provide a comprehensive update of the topic plantar fasciitis focused in the soccer players. "
"After reviewing the manuscripts included in this work, surprise that there is no data in which the Silfverskiöld test is performed, this test explores the passive mobility of the ankle and the degree of dorsiflexion in the supine position."
"The shortage of reported publications in soccer may mask other PF-associated injuries"
2.- Keywords: Please add "systematic review"
Thanks, it has been added
3.- Introduction: well-reviewed. I still see some track changes.
Methods: Well described. The review has adhered the guidelines for conducting a systematic review, such as PRISMA. However, since relevant papers were nearly non-existent, I understand why the manuscript has no table that shows the primary studies with their study design, country, sample, study population, analysis and findings, etc., which is the norm in a systematic review method.
Thank you very much. Indeed, as you indicated, as there were only 3 manuscripts, we thought that it was not necessary to create a table.
4.- Results and discussion: fairly written considering the limited relevant studies. The authors have reported that "there were almost nonexistent number of articles. Although we suspect that many soccer players suffer from this pathology, we believe they are not specifically treated and therefore no conclusive studies have been carried out in this area". But what are the implications of non-diagnoses? That is not clear in the conclusion.
Thank you for this remark. True, but with such a small number of publications, it is difficult for us to expand on the conclusion, without made guessing. In fact, we have already stated that “although we suspect that many soccer players suffer from this pathology, we believe they are not specifically treated and therefore no conclusive studies have been carried out in this area”.
Following your suggestion, at the end of the conclusions we have introduced the following sentence: “The shortage of reported publications in soccer may mask other PF-associated injuries.”
Reviewer 3 Report
The article entitled "Plantar fasciitis in soccer players. Systemic review" tries to find relevant aspects of the Plantar Fasciitis (PF) in soccer players. However, most of the content focus in the pathology of PF in different sports and its etiology. Therefore, the scientific contribution of the research is not clear.
The abstract does not describe the objective of the research and does not present its main contribution. Please rewrite the abstract considering the whole structure or content of the research. All abbreviations must be defined when they are mentioned within the text for the first time, PF.
The introduction section is too long (more than half of the content of the manuscript) and does not explain the relationship between PF and musculoskeletal injures in soccer players. The information presented in the introduction section can be found already published in different papers. Therefore, it is not clear the contribution of the research.
Add the reference in Figure 1 and include the labels a) and b).
Line 136, Soccer is not the most practiced sport in the world. Although the lower limbs are the most affected parts in this sport, there are regions of the body predisposed to suffer a musculoskeletal alteration in relation to the sport practiced, revise https://doi.org/10.1016/j.ft.2019.08.002
Which was the period of time considered in the review? this period must be included in the text. The Assessment of risk of bias is missing. Furthermore, the selection criteria for the analysis of the articles is not mentioned.
Add the title of Figure 2 and modify/correct the image. The search databases do not agree with the information within the text, please correct. It seems that the methodology was not conducted correctly.
All Figures must be referred within the text (Figure 2).
The references of the 8 articles found must be included in the results section, 5 general reviews and 3 PF in soccer sport. Therefore, the reader can easily identify the articles of the classification.
The information presented in the manuscript does not contribute with new knowledge in the understanding of the PF in soccer players. The biomechanics of the foot in soccer players should be described and related with the etiology of the PF. How can be related the biomechanics of the soccer player with the pathology of the PF?
Lines 188-189, Add the references in the first sentence.
Rewrite the conclusion base on the research.
Author Response
Dear Reviewer,
Thank you for taking the time to read the manuscript and give us valuable comments. We tried to address all of them. We hope, that the corrected version meets your requirements and will be accepted for publication
1.- The article entitled "Plantar fasciitis in soccer players. Systemic review" tries to find relevant aspects of the Plantar Fasciitis (PF) in soccer players. However, most of the content focus in the pathology of PF in different sports and its etiology. Therefore, the scientific contribution of the research is not clear.
The abstract does not describe the objective of the research and does not present its main contribution. Please rewrite the abstract considering the whole structure or content of the research. All abbreviations must be defined when they are mentioned within the text for the first time, PF.
Thank you for your comment. Indeed, we have analysed plantar fasciitis also in other sports as well, given the limited number of publications in the field of football. In the summary and in the conclusions we have introduced an additional sentence which might clarify the situation a little more: “The shortage of reported publications in soccer may mask other PF-associated injuries.”
Also, in the summary we have introduced these other sentences: “The purpose of this review was to provide a comprehensive update of the topic “plantar fasciitis” focused in the soccer players”.
“After reviewing the manuscripts included in this work, surprise that there is no data in which the Silfverskiöld test is performed, this test explores the passive mobility of the ankle and the degree of dorsiflexion in the supine position”.
Concerning abbreviations, we have completed information
2.- The introduction section is too long (more than half of the content of the manuscript) and does not explain the relationship between PF and musculoskeletal injures in soccer players. The information presented in the introduction section can be found already published in different papers. Therefore, it is not clear the contribution of the research.
Thank you for this remark. It is true that the introduction may be a little long, but we try to contextualise well that we are going to analyse. Also, you are right that there is information that could be obtained in other articles, but as we say, we intend to give a clear and broad view of the issue.
3.- Add the reference in Figure 1 and include the labels a) and b).
Thanks. We have added the text of the figure in English. We have also introduced the two sections that you have mentioned
4.- Line 136, Soccer is not the most practiced sport in the world. Although the lower limbs are the most affected parts in this sport, there are regions of the body predisposed to suffer a musculoskeletal alteration in relation to the sport practiced, revise https://doi.org/10.1016/j.ft.2019.08.002
Thank you very much for the information that you provide. The article is certainly interesting, however, we consider that more general aspects are analysed and that this would lead to have a more extended introduction even further. From the point of view of the discussion, it is far from the objective of this review.
5.- Which was the period of time considered in the review? this period must be included in the text. The Assessment of risk of bias is missing. Furthermore, the selection criteria for the analysis of the articles is not mentioned.
Thank you for this remark. We did not set a time period, we considered everything that was published in English. As for the risk of bias assessment, described by Cochrane in 2008, that is more focused on randomised clinical trials, we thought that it was not necessary to apply in this review. In addition, there were only 3 articles concerning the objective of our work
6.- Add the title of Figure 2 and modify/correct the image. The search databases do not agree with the information within the text, please correct. It seems that the methodology was not conducted correctly. All Figures must be referred within the text (Figure 2).
Thanks. It has been corrected
7.- The references of the 8 articles found must be included in the results section, 5 general reviews and 3 PF in soccer sport. Therefore, the reader can easily identify the articles of the classification.
Thank you very much for your appreciation. We thought that along of the discussion we have included these specific articles concerning soccer.
8.- The information presented in the manuscript does not contribute with new knowledge in the understanding of the PF in soccer players. The biomechanics of the foot in soccer players should be described and related with the etiology of the PF. How can be related the biomechanics of the soccer player with the pathology of the PF?
Thank you for this remark. In the manuscript presented by Petraglia et al (reference 1 in our paper), the authors explain extensively the causes of plantar fasciitis, by this we have considered that it was exposed by these authors could be in a repetitive way. On the other hand, with respect the biomechanics of gait, soccer players do not differ from other people. In fact, in the introduction we have included a section "Plantar fascia and the Windlass mechanism", focuses on the role of the fascia in the movement of the foot.
If we include more information about biomechanics, as you mentioned, the introduction would be too long, and on the other hand, these aspects can be reviewed in more general articles. It is in the same line that you mentioned before.
9.- Lines 188-189, Add the references in the first sentence.
Thanks. The reference has been added
10.- Rewrite the conclusion base on the research.
Following your suggestion, at the end of the conclusions we have introduced the following sentence: “The shortage of reported publications in soccer may mask other PF-associated injuries.”
Round 2
Reviewer 1 Report
The authors have addressed all of the major concerns. I think the overall research study can still be improved (mostly agree with other reviewers).
Author Response
Thank you very much for your comments. In this version we try to explain some of the aspects that may have been a bit weak in the explanations and approaches.
Reviewer 3 Report
How can you conclude that the soccer player suffers pain in the sole of the foot compatible with PF? Where is the evidence or information that supports this idea? Especially when there is almost nonexistent number of articles.
A reference should be added in Figure 1.
Although the authors made some changes in the manuscript, there are still some critical points that were not completely addressed. A paper is acceptable only if there is convincing evidence and it also adds a new and important result to the field. I regret that it is not possible to send you a more favorable report on this manuscript.
Author Response
- How can one conclude that the soccer player suffers from pain in the sole of the foot compatible with PF? Where is the evidence or information to support this idea? Especially when the number of articles is almost non-existent.
Indeed there is no scientific evidence for the above given the small number of publications. We have inserted the following two paragraphs at the end, prior to the conclusions
Based on our clinical experience, both in the normal population and in other sports (mainly runners on foot), the plantar fascia tends to lose elasticity. To this we could add that we have also observed that it is more frequent in people who do less exercise, and more in those who do not stretch the fascia. A fact that we have already published (5) and which shows that this situation is the opposite of the preventive mechanism.
In soccer players, the most common technical movements are tackling, being tackled, sprinting, running, turning, jumping, landing and kicking the ball. These are all complex movement patterns that require a high degree of coordination and body control. The biomechanics and explosiveness of the stride, as well as the hyperextension of the ankle in the kicking mechanics generate a risk situation on the plantar fascia that favors its injury. In this regard, Laurie et al, showed that the comparison of different studies is difficult due to the different classification systems used to indicate the severity of the injury; some studies are classified according to the medical treatment used, while others on the days of absence from competition (42).
- A reference should be added to Figure 1.
With respect the Figure 1 is self-composed by taking several images and composing the figure as my own.